# Self Manipulated Cervical Spine Leads to Posterior Disc Herniation and Spinal Stenosis

**DOI:** 10.3390/brainsci9060125

**Published:** 2019-05-29

**Authors:** Wyatt McGilvery, Marc Eastin, Anish Sen, Maciej Witkos

**Affiliations:** 1Department of Emergency Medicine, Loma Linda University Medical Center, Loma Lind, CA 92354, USA; mwitkos@llu.edu; 2Department of Neurosurgery, Loma Linda University Medical Center, Loma Lind, CA 92354, USA; meastin@llu.edu (M.E.); asen@llu.edu (A.S.)

**Keywords:** cervical disk herniation, spinal stenosis, neurosurgery, discectomy, arthrodesis, anterior approach, self manipulation, cervical spine, spinal manipulation therapy, acute trauma

## Abstract

The authors report a case in which a 38-year-old male who presented himself to the emergency department with a chief complaint of cervical neck pain and paresthesia radiating from the right pectoral region down his distal right arm following self-manipulation of the patient’s own cervical vertebrae. Initial emergency department imaging via cervical x-ray and magnetic resonance imaging (MRI) without contrast revealed no cervical fractures; however, there was evidence of an acute cervical disc herniation (C3–C7) with severe herniation and spinal stenosis located at C5–C6. Immediate discectomy at C5–C6 and anterior arthrodesis was conducted in order to decompress the cervical spinal cord. Acute traumatic cervical disc herniation is rare in comparison to disc herniation due to the chronic degradation of the posterior annulus fibrosus and nucleus pulposus. Traumatic cervical hernias usually arise due to a very large external force causing hyperflexion or hyperextension of the cervical vertebrae. However, there have been reports of cervical injury arising from cervical spinal manipulation therapy (SMT) where a licensed professional applies a rotary force component. This can be concerning, considering that 12 million Americans receive SMT annually (Powell, F.C.; Hanigan, W.C.; Olivero, W.C. A risk/benefit analysis of spinal manipulation therapy for relief of lumbar or cervical pain. *Neurosurgery*
**1993**, *33*, 73–79.). This case study involved an individual who was able to apply enough rotary force to his own cervical vertebrae, causing severe neurological damage requiring surgical intervention. Individuals with neck pain should be advised of the complications of SMT, and provided with alternative treatment methods, especially if one is willing to self manipulate.

## 1. Introduction

The etiology of cervical herniated nucleus pulposus, or herniated discs, most often arise due to age related degenerative properties such as dehydration and weakening of the posterior annulus fibrosus and nucleus pulposus, ultimately leading to posterior disc herniation and protrusion into the spinal canal. A lesser common etiology of cervical herniated nucleus is traumatic, usually associated with a high energy external force resulting in extreme hyperflexion or hyperextension of the cervical vertebrae [1]. However, we are currently unaware of any other case where self manipulation (forced external rotation) of one’s own cervical vertebrae has led to traumatic posterior cervical disc protrusion and severe central spinal stenosis requiring immediate surgical intervention. This case study has been submitted to, and approved by Loma Linda University Health’s Institutional Review Board (IRB) in order to ensure all ethical criterion has been met.

## 2. Case Report 

A 38-year-old male with a remote history penetrating neck trauma in 2013 presented himself to the emergency department with a chief complaint of acute posterior cervical neck pain and paresthesia. Prior to the onset of acute neck pain, the patient had attempted to manipulate his own cervical vertebrae in order to relieve himself of a continuously nagging “crick in the neck.” After his first attempt of using both hands (one hand on the anterolateral aspect of his mandible, and the other hand on his occipital region) to apply external rotation and torsion of his cervical vertebrae, he heard a loud crunching sound, followed by immediate neck pain. This neck pain worsened with movement, associated with unilateral radiating pain and paresthesia down his right arm and extending to his right pectoral region, whole body numbness below his shoulders, and a throbbing component of pain when at rest. Patient denied any other neurological symptoms including headache, nausea, vomiting, blurry vision, changes in hearing, loss of balance, aphasia, loss of extremity strength, fecal or urinary incontinence or retention, and denies any use of aspirin or any other anticoagulation medications. 

Physical examination found that the patient had 5/5 strength throughout except a 4+/5 distal right hand strength. The patient also had decreased sensation to light touch below the neck, which was worse on the right side. The patient did not have clonus or Hoffman’s sign, no Babinski’s sign, nor hyperreflexia. Initial MRI results showed a large 5–6 mm traumatic right paracentral posterior disc protrusion with disruption of the posterior disc annulus at the C5–C6 level with associated severe central spinal stenosis and ligamentous damage to the posterior longitudinal ligament (PLL). There was also mild to moderate central spinal stenosis secondary to smaller disc protrusions at the C3–C4 and C6–C7 levels (Figure 1). Initial MRI showed no osseous fractures of the vertebrae, nor asymmetry in the alignment of any faucet joints. 

A discectomy and arthrodesis surgical intervention was opted to be performed on the patient due to myelopathy upon examination. In addition to this, the patient had radiculopathy and ligamentous injury. 

## 3. Surgical Interventions 

Given the patient’s presenting symptoms and neurological deficits (hand weakness, numbness), combined with the MRI findings of an acute disc herniation with severe spinal cord compression, neurosurgery was consulted. As the patient had acute neurological changes, it was recommended that the patient undergo urgent surgery for an anterior C5–C6 discectomy and arthrodesis.

After the patient was placed under general anesthesia, electrodes were placed for motor-evoked potential and somatosensory-evoked potentials. Baseline potentials were then obtained and remained stable throughout the entire procedure. A transverse right anterior neck incision was made at the C5–C6 level, confirmed by intraoperative X-ray. Once appropriate anatomical structures were divided and retracted, a large disc fragment was encountered between C5–C6 and removed below the PLL. Decompression was then confirmed with a nerve hook. A 6 mm cadaveric structural allograft was then sized and positioned with X-ray. Meticulous disc space hemostasis was obtained. 

Copious irrigation was then performed prior to a 12 mm anterior cervical plate being secured with 4.0 ×16 mm screws in C5 and variable screws in C6. 

The patient tolerated the procedure well and intraoperative X-rays confirmed proper placement of the plate screws and interbody device. The incision was then irrigated with antibiotic irrigation and closed in a layered fashion. The patient was then extubated and transported to postoperative recovery in a stable condition.

A postoperative CT scan confirmed the proper anterior fixation of the plate screws, interbody device, and bone graft placement following the patient’s discectomy and arthrodesis (Figure 2). Postoperative anterior and lateral X-rays (Figure 3b,a respectively) were also obtained in order to visualize and confirm proper placement of the anterior cervical plate and evaluation of alignment.

On follow up at one month, the patient had resolution of his numbness and paresthesia, however, he continued to note posterior midline neck pain. He reported cannabinoid and methamphetamine use despite discussion to avoid drugs while fusion was occurring. He was subsequently lost to follow up.

## 4. Discussion

When researching other published literature for similar cases regarding cervical manipulation resulting in cervical disc herniation and subsequent spinal stenosis, few reports were found. However, publications like that by Yang [2] indicate the possibilities of cervical spinal stenosis that can arise from cervical SMT. The most common injuries associated with cervical SMT all seem to be reported occurring at the C5–C6 level. 

It is our belief that patients should be advised of alternative therapy methods such as physiotherapy when discussing the treatment of acute cervical neck pain due to the unusually high risk/benefit ratio of cervical SMT^4^.

Our case study is relatively unique, as there have been no other reports to our knowledge of the self-manipulation of one’s own cervical spine leading to severe acute disc herniation and neurological deficits. Most reports of this sort of injury have been attested due to age related degenerative factors, and a few cervical SMTs provided by other licensed professionals. In addition to this, it is known that 85% of all patients suffering from symptoms of an acute herniated disc will resolve themselves within 8–12 days without the need of specific treatment [3]. The fact that our patient’s pathology included a severe herniation causing spinal stenosis with neurological deficits in need of urgent neurosurgical intervention further attests to the uniqueness of this case. 

## 5. Conclusions

Self manipulation of the cervical vertebrae may result in acute traumatic cervical disc herniation, severe enough to result in focal neurological deficits. Individuals who feel the need to stretch their neck in an external torsion or rotary manner should be educated in the possible dangers of doing so. Individuals presenting with acute neck or arm pain, leg pain, paresthesia, or other neurological symptomatology post self cervical manipulation or “stretching” should be evaluated for possible cervical disc protrusions and spinal stenosis via MRI.

## Figures and Tables

**Figure 1 brainsci-09-00125-f001:**
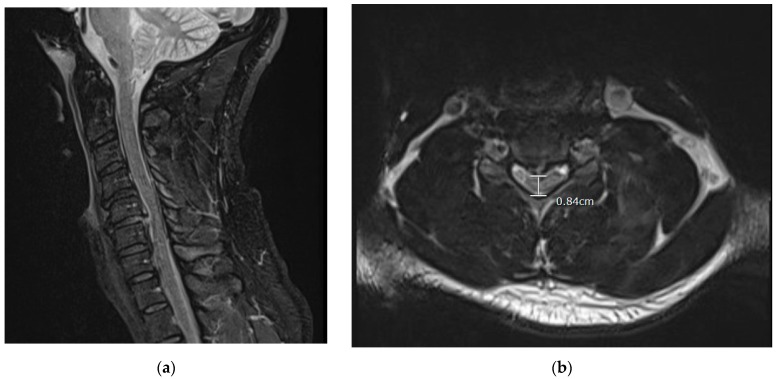
Preoperative MRI (STIR sequence sagittal (Figure 1a) and T2 axial (Figure 1b)) taken in the emergency department shows the pathology of C5–C6 posterior acute cervical disc herniation with increased signal in the posterior longitudinal ligament and severe spinal cord compression with cord signal change status post self-manipulation of neck. The encircled area also shows the protrusion of the posterior herniated disc causing spinal stenosis. MRI = magnetic resonance imaging.

**Figure 2 brainsci-09-00125-f002:**
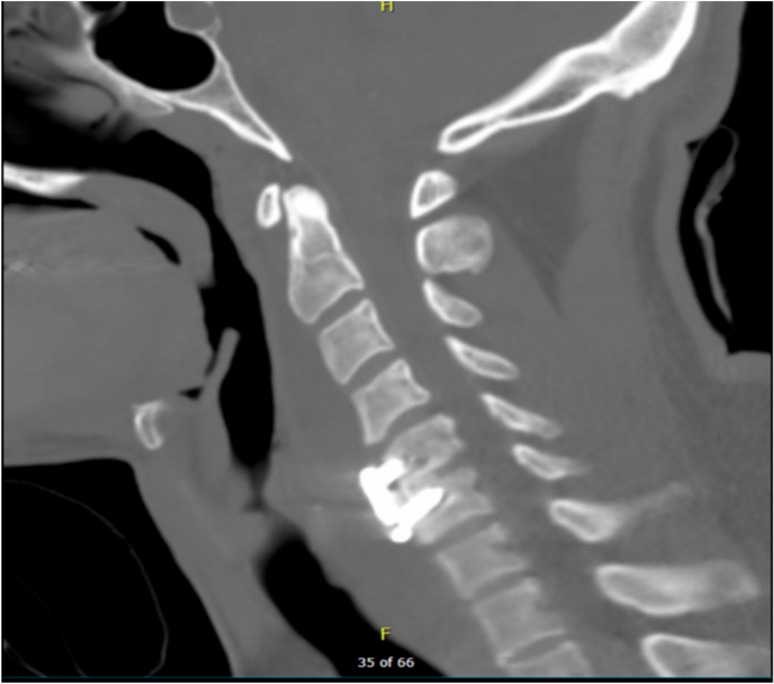
Postoperative CT scan showing spinal cord decompression following surgical discectomy and anterior arthrodesis of the C5–C6 vertebrae. CT scan = computerized tomography scan.

**Figure 3 brainsci-09-00125-f003:**
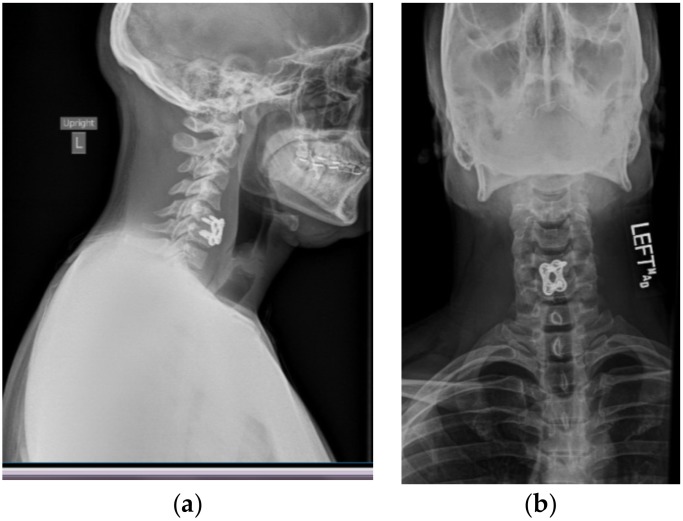
Postoperative X-ray showing postsurgical changes related to anterior fusion at C5–C6, with interval decrease in prevertebral soft tissue swelling. C1–C7 are visualized on the lateral view (Figure 3a) for evaluation of alignment. There is straightening of the normal cervical lordosis. Alignment is otherwise grossly unremarkable when allowing for patient rotation. Anterior instrumentation is noted at C5–C6 on both lateral and anterior (Figure 3b) views.

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
