# Peer review of "Self Manipulated Cervical Spine Leads to Posterior Disc Herniation and Spinal Stenosis"

_brainsci, 2019, doi:10.3390/brainsci9060125_

Reviewer 1 Report

The authors presented an interesting case of traumatic disc herniation from self manipulation of the cervical spine. I think this report is of interest to the readership. The authors should include post-operative xrays in the manuscript.

Author Response

Point 1: The authors presented an interesting case of traumatic disc herniation from self manipulation of the cervical spine. I think this report is of interest to the readership. The authors should include post-operative xrays in the manuscript.

Response 1: I will try to obtain any postoperative x-ray images. However, post-op, the patient was uncooperative with further treatment, and denied many other CT scans, x-rays, and other diagnostic measures. 

Reviewer 2 Report

The authors present an interesting case study that is worth a discussion and presentation to the community.

I think it would be interesting to comment on whether any dislocation of the facet Joints was seen on MRI.

You are describing that no fracture was seen on x-ray and MRI. However, CT is the goldstandard for the detection of cervical spine fracture. It has to be taken to account that a small fracture could have been missed on the initial images.

A comment on whether a myelopathy was seen on MRI would be an important Information.

To my knowledge, an 4+/5 motoric decrease is not an indication for urgent surgery. Can you comment on that. Because you are saying that most of the disc herniations improve after 2 weeks. Why did you choose to make urgent surgery and not wait for a few days?

Figure 2: the first sagittal image is a T2w Image and not a STIR Image. STIR has no signficance in the assessment of spinal canal stenosis.

I would remove the measurements of the width of the spinal canal in Figure 2, because it is obvious to see that the stenosis is severe. The severity of the stenosis grade is not measured typically.

Did you take into account that the disc herniation might have already been there before the trauma and only worsened during the traumatic event?!

Author Response

Point 1: I think it would be interesting to comment on whether any dislocation of the facet Joints was seen on MRI.
You are describing that no fracture was seen on x-ray and MRI. However, CT is the goldstandard for the detection of cervical spine fracture. It has to be taken to account that a small fracture could have been missed on the initial images.
Response 1: There is no asymmetry in the alignment of the facet joints. The MRI shows ligamentous injury to the PLL. Even if there was a small fracture that was not detected on the MRI it would not be enough to cause ligamentous injury and would not contribute to the patients pathology. Even though CT is the gold standard fractures are still detected on MRI to the careful observer.
Point 2: A comment on whether a myelopathy was seen on MRI would be an important Information.
To my knowledge, an 4+/5 motoric decrease is not an indication for urgent surgery. Can you comment on that. Because you are saying that most of the disc herniation improve after 2 weeks. Why did you choose to make urgent surgery and not wait for a few days?
Response 2: We elected to perform surgery because our pt had myelopathy on exam. He also had radiculopathy, and ligamentous injury. It was also determined that the patient would likely be lost to follow up which would make conservative management a higher risk to the patient.  
Point 3: I would remove the measurements of the width of the spinal canal in Figure 2, because it is obvious to see that the stenosis is severe. The severity of the stenosis grade is not measured typically.

Response 3: We can remove the measurements if that is what is recommended for publication.
Point 4: Did you take into account that the disc herniation might have already been there before the trauma and only worsened during the traumatic event?!”
Response 4: It is possible that he had a disc herniation before the traumatic event. However, he had not previously experienced any symptoms before the event if there was a herniation before the event it would be likely that he would likely experience some symptoms before the event.